# SingNet: Towards a Large-Scale, Diverse, and In-the-Wild Singing Voice Dataset

## Abstract

The lack of a publicly-available large-scale and diverse dataset has long been a significant bottleneck for singing voice applications like Singing Voice Synthesis (SVS) and Singing Voice Conversion (SVC). To tackle this problem, we present SingNet, an extensive, diverse, and in-the-wild singing voice dataset. Specifically, we propose a data processing pipeline to extract ready-to-use training data from sample packs and songs on the internet, forming 3000 hours of singing voices in various languages and styles. Furthermore, to facilitate the use and demonstrate the effectiveness of SingNet, we pre-train and open-source various state-of-the-art (SOTA) models on Wav2vec2, BigVGAN, and NSF-HiFiGAN based on our collected singing voice data. We also conduct benchmark experiments on Automatic Lyric Transcription (ALT), Neural Vocoder, and Singing Voice Conversion (SVC). Audio demos are available at: `https://singnet-dataset.github.io/`.

## 1 Introduction

Singing Voice Synthesis (Liu et al., 2022a; Zhao et al., 2024) and Conversion (Liu et al., 2021; Zhang et al., 2023) have attracted much attention from industry and academic communities due to their business value in the entertainment and music industry. As illustrated in Shi et al. (2024), high-quality, extensive, and diverse singing voices are essential to these applications but are always lacking due to the high cost of data acquisition (e.g., professional singers, recording environments, etc.). To tackle this issue, some data scaling methods are proposed, including web crawling (Ren et al., 2020) and data augmentation (Guo et al., 2022), but are often limited in quality and quantity (Shi et al., 2024). More recently, ACESinger (Shi et al., 2024) tried to generate extensive singing voices via commercial AI singers in ACEStudio. [1] However, to create high-quality singing voices via such a method, many professional producers are required to tune the in-detailed pitch, phoneme, and duration information for different songs and singers, making it manpower-consuming and inconvenient for scaling up.

The power of data scaling has been proven effective in similar applications like speech generation (He et al., 2024). The Emilia (He et al., 2024) dataset was recently proposed for in-the-wild speech data scaling up with an open-sourced data processing pipeline. It collected 101k hours of data from various sources and achieved considerable results in Text-to-Speech (TTS). Inspired by Emilia (He et al., 2024), this study utilizes the massive in-the-wild singing data from multiple sources. Specifically, we propose a data processing pipeline to extract ready-to-use training data via state-of-the-art (SOTA) deep learning methods (Cooper et al., 2022; Cuesta et al., 2020; Solovyev et al., 2023; Fabbro et al., 2024; Tang et al., 2024), Digital Signal Processing (DSP) algorithms (McFee et al., 2015; Openvpi, 2022), and Virtual Studio Technology (VST) plugins. We collect 2629 and 321 hours of singing data from in-the-wild songs and sample packs [2] on the internet, respectively, forming a multilingual and multi-style dataset with around 3000 hours of singing data. To facilitate the use and illustrate the effectiveness of SingNet, we pre-train and open-source various SOTA checkpoints based on the data we collected, including Wav2vec2 (Baevski et al., 2020), BigVGAN (Lee et al., 2023), and NSF-HiFiGAN (Liu et al., 2022a) models. [3] We also conduct benchmark experiments on Automatic Lyric Transcription (ALT), Neural Vocoder, and Singing Voice Conversion (SVC).

---

[1] `https://acestudio.ai/`

[2] Sample pack is a collection of audio samples that music producers can use in their songs, containing ready-to-use high-quality vocal stems recorded by professional singers.

[3] We are committed to make these checkpoints publicly available after the double-blind review period.

Table 1: A comparison of SingNet with existing singing voice datasets. "SR" means Studio Recording, "SS" means Sample Pack, "SP" means Source Separation, "MIS" means uncoded Indigenous languages, and "*" means extensibility, which features an automatic pipeline for efficiently further scaling up. Datasets are sorted by the release year. Compared with existing datasets, SingNet is the largest, with extensibility and more diverse styles and languages.

| Dataset | Data Source | Dur. (hour) | Style | Lang. | Samp. Rate (Hz) |
|---|---|---|---|---|---|
| NUS-48E (Duan et al., 2013) | SR | 2.8 | Children/Pop | ZH | 44.1k |
| Opera (Black et al., 2014) | SR | 2.6 | Opera | IT/ZH | 44.1k |
| VocalSet (Wilkins et al., 2018) | SR | 8.8 | Opera | EN | 44.1k |
| CSD (Choi et al., 2020) | SR | 4.6 | Children | EN/KO | 44.1k |
| PJS (Koguchi et al., 2020) | SR | 0.5 | Pop | JA | 48k |
| NHSS (Sharma et al., 2021) | SR | 4.1 | Pop | EN | 48k |
| OpenSinger (Huang et al., 2021) | SR | 51.8 | Pop | ZH | 44.1k |
| Kiritan (Ogawa & Morise, 2021) | SR | 1.2 | Pop | JA | 96k |
| KiSing (Shi et al., 2022) | SR | 0.9 | Pop | ZH | 44.1k |
| PopCS (Liu et al., 2022a) | SR | 5.9 | Pop | ZH | 44.1k |
| M4Singer (Zhang et al., 2022) | SR | 29.7 | Pop | ZH | 48k |
| PopBuTFy (Liu et al., 2022b) | SR | 30.7 | Pop | EN | 44.1k |
| Opencpop (Wang et al., 2022) | SR | 5.2 | Pop | ZH | 44.1k |
| SingStyle111 (Dai et al., 2023) | SR | 12.8 | Children/Folk/Jazz Opera/Pop/Rock | EN/IT/ZH | 44.1k |
| GOAT (Zheng et al., 2024) | SR | 4.5 | Opera | ZH | 48k |
| ACESinger (Shi et al., 2024) | SVS | 321.8 | Pop | EN/ZH | 48k |
| SingNet-SS* | In-the-wild | 2629.1 | ACG/Classical/EDM Folk/Indie/Jazz Light/Pop/Rap/Rock | DE/ES/EN/FR IT/JA/KO/RU ZH-YUE/ZH | 44.1k |
| SingNet-SP* | In-the-wild | 334.3 | EDM/Folk/Jazz Opera/Pop/Rap | AR/DE/ES/EN FR/ID/PT/RU ZH/MIS | 44.1k |

The main contributions of this paper are summarized as follows:

- We introduce *the first open-source data processing pipeline* to automatically extract ready-to-use singing voice training data from songs and sample packs on the internet, with the help of SOTA deep learning, DSP, and VST technologies.

- With the pipeline, we present SingNet, a large-scale, diverse, and in-the-wide dataset for singing voice applications. SingNet can be extended dynamically over time by applying the data processing pipeline to more sources. *To the best of our knowledge, this is the largest in-the-wild singing voice dataset to date*, as presented in Table. 1.

- To facilitate the use and illustrate the effectiveness of SingNet, we pre-train and open-source SOTA Wav2vec2, BigVGAN, and NSF-HiFiGAN checkpoints based on our collected data. We also conducted benchmark experiments on ALT, Neural Vocoder, and SVC.

## 2 RELATED WORK

This section reviews the existing singing voice datasets and introduces the development of ALT, Neural Vocoder, and SVC, explaining how our collected large-scale data can benefit these tasks.

### 2.1 SINGING VOICE DATASETS

Singing Voice Datasets are always scarce due to the high recording and annotation costs. The MIR-1K (Hsu & Jang, 2010) dataset establishes the first comprehensive dataset for singing voice separation. Since then, many datasets have been constructed similarly in recent years, as illustrated in Table. 1. Regarding these studio-recorded datasets, it can be observed that (1) Most datasets have limited data scales, ranging from 0.5 to 51.8 hours; (2) Most datasets are limited to Pop Songs, with only a few focusing on other styles; (3) Most datasets are limited to Chinese Singing, with only a few focusing on other languages. Recently, ACESinger (Shi et al., 2024) has been proposed to tackle the data scale issue using commercial SVS technologies. However, generating training data with such a method is manpower-consuming for scaling up. *In response to these limitations, this paper introduces the first open-source data processing pipeline on massive in-the-wild data from the internet, forming SingNet, a 3000-hour singing voice dataset with various languages, singers, and styles.*

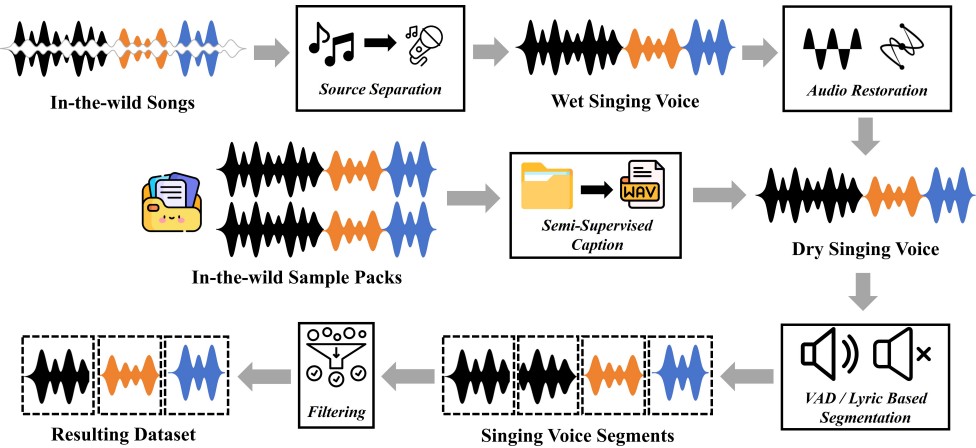

Figure 1: An overview of the SingNet data processing pipeline. It processes in-the-wild songs and sample packs into a ready-to-use dataset for model training.

## 2.2 AUTOMATIC LYRIC TRANSCRIPTION

ALT aims to extract lyrics from a singing voice signal. Following the advancement in Automatic Speech Recognition (ASR) with Self-Supervised Learning (SSL) (Baevski et al., 2020; Hsu et al., 2021; Qian et al., 2022), recent ALT works are also trying to adapt SSL models on singing voices. Specifically, Ou et al. (2022) successfully adapted Wav2vec2 embeddings for ALT via transfer learning, marking a significant leap in model performance. Zhuo et al. (2023) leverages Whisper (Radford et al., 2023) and ChatGPT (Achiam et al., 2023) post-processing to further reduce error rates. However, due to the lack of large-scale singing voice data, these works heavily rely on fine-tuning and transfer learning from speech-pre-trained SSL models via various techniques, which is inconvenient. In this paper, we pre-trained an SSL model, Wav2vec2 (Baevski et al., 2020), based on our collected large-scale singing voice. We conducted experiments to show that our pre-trained model can be directly adapted on ALT and perform similarly compared with Ou et al. (2022).

## 2.3 NEURAL VOCODER

The vocoder aims to convert waveform from an acoustic feature outputted by the acoustic model. Among different types of vocoders, the neural network-based ones (van den Oord et al., 2016; Kalchbrenner et al., 2018; Prenger et al., 2019; Su et al., 2020; Kong et al., 2021; Lee et al., 2023) are essential due to their superior synthesis quality compared to the DSP-based ones (Kawahara, 2006; Morise et al., 2016). High-quality, extensive, and diverse training data are crucial to the vocoder's model performance. Specifically, BigVGAN (Lee et al., 2023) adapts large-scale speech and general sound data mixture with additional losses (Gu et al., 2024b;a), obtaining SOTA performance on speech and audio effects. Meanwhile, Openvpi (2024) utilizes an extensive collection of studio-recorded singing voice data, resulting in SOTA performance on the singing voice. In this paper, we conduct vocoder pre-training on our collected large-scale data, providing SOTA open-sourced checkpoints and experimental benchmarks.

## 2.4 SINGING VOICE CONVERSION

SVC aims to transform a singing signal into the voice of a target singer while maintaining the original lyrics and melody (Huang et al., 2023). Current SVC systems usually decouple the input features into two parts: the speaker agnostic and specific representations. The semantic-based features from pre-trained models (Qian et al., 2022; Radford et al., 2023; Zhang et al., 2023) are widely used as the speaker-agnostic representation (Liu et al., 2021). For speaker-specific representations, learnable speaker embeddings (Jiang et al., 2024; Shen et al., 2024; Ju et al., 2024), known as the zero-shot technique, have been proposed recently, making it possible to utilize large-scale in-the-wild data without speaker annotations. This paper uses these recent zero-shot SVC models (Chen et al., 2024; Wang et al., 2024) to build singing voice generation benchmarks on different data scales.

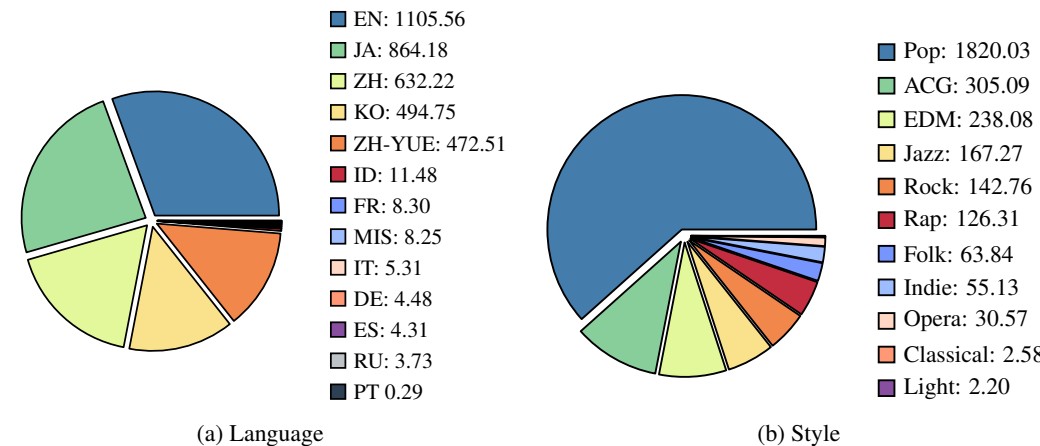

Figure 2: Duration statistics (hours) of SingNet by language and style sorted by the data scales. "MIS" means uncoded Indigenous languages

## 3  SINGNET AND ITS DATA PROCESSING PIPELINE

As discussed in Section 2, existing singing voice datasets are undiversified regarding styles and languages with limited data scales, which will restrict the performance of singing voice applications (Zhao et al., 2024). To address this limitation, we propose SingNet, an extensive, multilingual, and diverse singing voice dataset that utilizes massive amounts of data from the internet. This section provides the construction details, necessary statistics, and analysis of SingNet.

### 3.1  DATASET CONSTRUCTION

SingNet comprises two data sources: In-the-wild songs and sample packs. We extract dry [4] singing voices from songs using SOTA source separation and audio restoration techniques and music production sample packs using our proposed semi-supervised caption system, as illustrated in Figure 1. The two different data sources are denoted respectively as SingNet-SS and SingNet-SP.

#### 3.1.1  SINGNET-SS CONSTRUCTION

The raw data for SingNet-SS are sourced from online music streaming platforms, with annotations including user-labeled lyrics, language, and genres. The processing pipeline is described as follows:

**Source Separation**: We use the source separation technique to extract wet singing voices from songs for further processing. Specifically, we utilize the open-source library from Solovyev et al. (2023) and its pre-trained model MDX23 from Fabbro et al. (2024).

**Audio Restoration**: We use SOTA VST3 [5] plugins for audio restoration. To make the processing procedure compatible with Python code and command line usage, we use Reaper [6] as our Digital Audio Workstation (DAW) and its FX Chain for batch processing. The details are listed below:

- FabFilter-Pro Q3 [7]: A 20hz low cut with a 22000hz high cut to exclude noises.

- Waves Clarity Vx Pro [8]: Default preset with full ambiance reduction for denoising

- kHs Gate [9]: Default preset with -40 dB threshold for denoising.

---

[4]"Dry" means the unprocessed audio and "wet" means the processed audio with effects like reverb.
[5]https://www.steinberg.net/technology/
[6]https://www.reaper.fm/
[7]https://www.fabfilter.com/products/pro-q-3-equalizer-plug-in
[8]https://www.waves.com/plugins/clarity-vx-pro
[9]https://kilohearts.com/products/kilohearts_ultimate

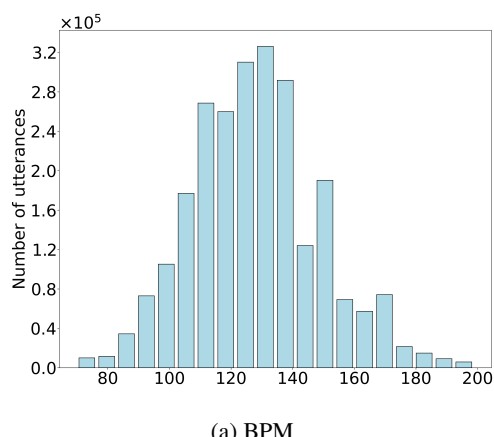
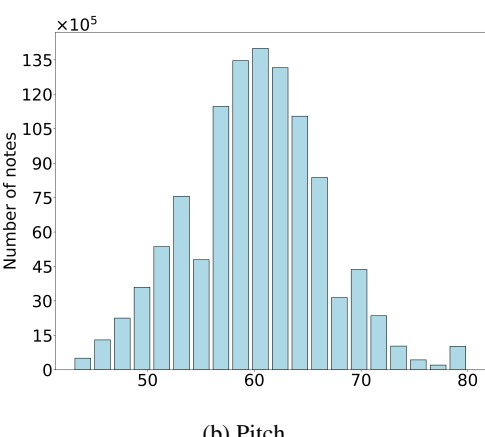

(a) BPM  (b) Pitch

Figure 3: BPM and Pitch statistics (occurrences) of SingNet. The pitch is illustrated as MIDI notes where A4=69=440Hz. Values outside of the illustrated ranges are considered errors and are removed .

- Waves Clarity Vx DeReverb Pro [10]: Singing 1 preset with double channel processing for removing reverberation.
- RX 10 Voice De-noise [11]: Default preset with music mode for denoising.
- RX 10 De-click [11]: Default preset to remove clicks.
- RX 10 De-plosive [11]: Default preset to remove pops and bumps.
- RX 10 Mouth De-click [11]: Default preset to remove saliva noise and lip smacks.
- FabFilter-Pro Q3 [7]: A 20hz low cut with a 22000hz high cut to exclude noises.

**Lyric-based Segmentation**: We apply segmentation according to the time stamps in the human-labeled lyrics. We trim the leading and trailing silences through Librosa (McFee et al., 2015).

### 3.1.2 SINGNET-SP CONSTRUCTION

The raw data for SingNet-SP are sourced from online sample pack libraries, including manually labeled genres and language annotations. The processing pipeline is described as follows:

**Semi-supervised Caption**: Music production sample packs contain singing voices and sounds from instruments and synthesizers. We build a semi-supervised caption system to extract dry singing voices from them. Specifically, we built a web application for human labeling and manually labeled samples from 768 sample packs into dry/wet sounds with 4 sample categories by people with academic and music production backgrounds. Then, we use these captions to train an audio classification model for further scaling up. The details are introduced in Appendix A. The examples of the four sample categories can be listed on our demo page [12], and their short definitions are listed as follows:

- **Acapella**: The leading singing voice stem.
- **Adlibs**: Short vocal melodies accompanying the leading voice.
- **Elements**: Human-edited vocal chops accompanying the instrument.
- **FX**: Long vocal chants padding the whole song.

**VAD-based Segmentation**: We apply segmentation through DSP-based Voice Activity Detection (VAD) from Openvpi (2022) and discard clips shorter than 0.5 seconds.

---

[10]https://www.waves.com/plugins/clarity-vx-dereverb-pro
[11]https://www.izotope.com/en/products/rx.html
[12]https://singnet-dataset.github.io/

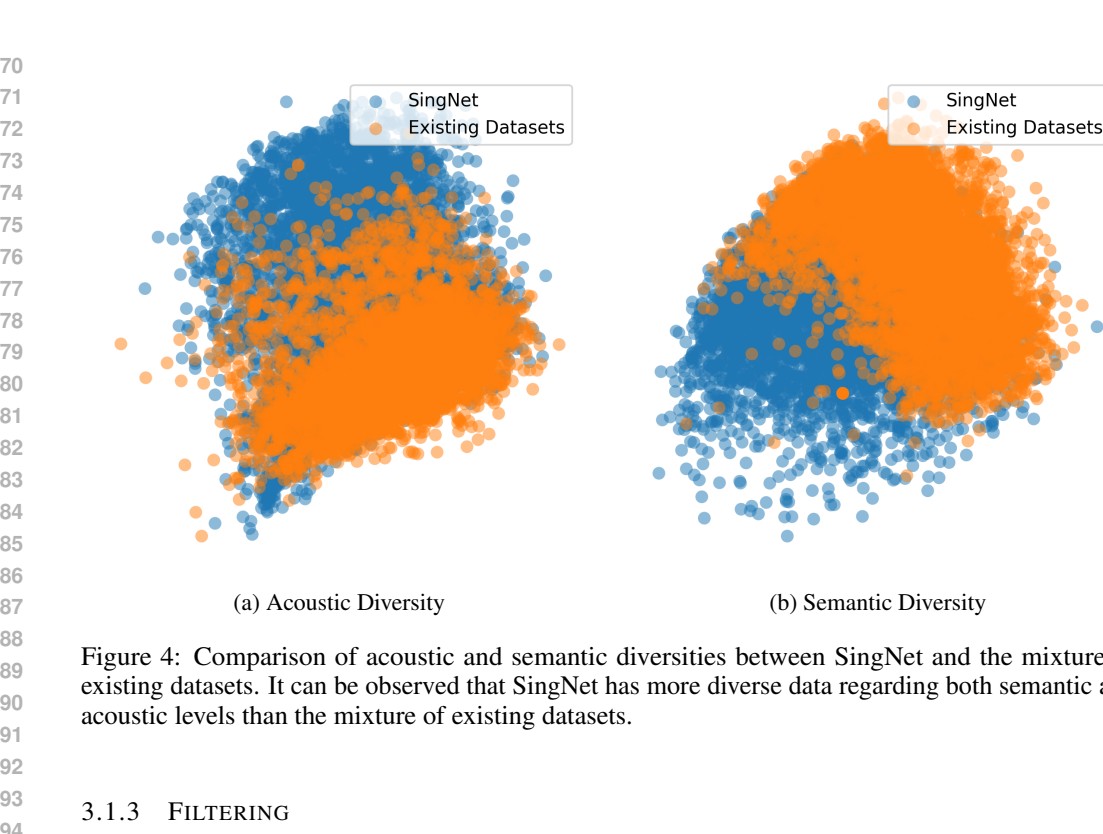

(a) Acoustic Diversity          (b) Semantic Diversity

Figure 4: Comparison of acoustic and semantic diversities between SingNet and the mixture of existing datasets. It can be observed that SingNet has more diverse data regarding both semantic and acoustic levels than the mixture of existing datasets.

### 3.1.3 FILTERING

To make our data compatible with existing singing voice models, we utilize multi-F0 detection proposed in Cuesta et al. (2020) to detect and exclude utterances with multiple singers singing simultaneously. Furthermore, source separation and audio restoration may not effectively handle all instrumental sounds and reverberation. Thus, the resulting singing voice data may be of low quality. To filter out these unwanted data, we fine-tune a singing voice scorer using the method in Cooper et al. (2022) on the SingMOS dataset (Tang et al., 2024) and apply the model on all audio segments, preserving only the singing voice data with a score higher than 3.0.

### 3.2 DATASET STATISTICS

The statistical results on language and style durations, pitch, and beats per minute (BPM) are illustrated in Fig. 2 and Fig. 3. We use librosa (McFee et al., 2015) to extract the BPM information and parselmouth (Jadoul et al., 2018) to extract the pitch information. Following Wang et al. (2022), the pitch is illustrated as MIDI notes where A4=69=440Hz.

It can be observed that:

- For language distribution, most utterances are distributed within English, Japanese, Chinese, and Korean, with a small number of minor languages like French and German. Notably, 8 hours of Indigenous Language exist, ranging from South American and Indian tribal sounds to Scandinavian and Irish folk music.

- For style distribution, most songs are Pop music, with a considerable amount of ACG, EDM, Jazz, Rock, and Rap songs. Some niche styles, like Folk, Indie, Opera, Classical, and Light music, exist with relatively smaller percentages.

- For BPM distribution, most utterances are distributed between 100 and 160. Since SingNet consists of soothing Bel Canto and high-speed EDM (like Speed Core) songs, utterances with lower and higher BPMs also exist, further contributing to the dataset's diversity.

- For pitch distribution, most notes are distributed between Note 50 (D3, 147 Hz) and Note 70 (B4, 494 Hz). Since SingNet consists of utterances from Opera and Tribal Chants, pitch notes in the lower and upper ends also exist, contributing to the dataset diversity.

### 3.3 DATASET ANALYSIS

SingNet comprises a collection of in-the-wild singing voice data with diverse styles, languages, singers, and recording environments. We compare its acoustic and semantic feature space to quantify this diversity with all existing singing voice datasets.

We randomly selected 5000 samples from SingNet. We choose different amounts of samples from each existing dataset according to their sizes to form a 5000-sample data mixture. To analyze the diversity of acoustic features, we leverage a pre-trained Mert model (Li et al., 2024)[13] to extract acoustic representations (the 12-th layer is used), capturing a variety of acoustic characteristics such as timbre, style, etc. For the semantic diversity analysis, we employ a pre-trained W2v-BERT model (Chung et al., 2021)[14] to generate semantic representations (the last layer is used), capturing language, content, etc. We then apply the Principal Component Analysis (PCA) algorithm to reduce the dimensionality of these representations to two. As illustrated in Fig. 4, SingNet exhibits a broader dispersion than the data mixture obtained from all the existing datasets, indicating the richer acoustic and semantic characteristic coverage in SingNet.

## 4 EXPERIMENTS

In this section, we pre-train and open-source SOTA Wav2vec2, BigVGAN, and NSF-HiFiGAN models to facilitate the use and show the effectiveness of SingNet. We also conduct benchmark experiments on ALT, Neural Vocoder, and SVC for subsequent research.

### 4.1 EXPERIMENT SETUP

#### 4.1.1 DATASETS

We utilize all the existing singing voice datasets for training, as illustrated in Table. 1, resulting in a singing voice mixture of 3500 hours with various recording qualities, styles, singers, and languages. Two datasets are used for evaluation; we randomly sample 1 hour and 6 hours of multilingual, multi-singer, and multi-style audio from SingStyle111 and SingNet-SS to form the Studio Recording and In-the-wild evaluation sets, respectively.

#### 4.1.2 PREPROCESSING

We resample all the training data to 16kHz for SSL pre-training and ALT fine-tuning. For Vocoder and SVC, we resample all the training data to 44.1kHz. These data will then be converted to an STFT matrix with an fft size of 2048, hop length of 512, window length of 2048, fmin of 0, and fmax of 22050, which will later be transformed into a mel-spectrogram with 128 mel-filters. The mel-spectrogram is normalized in log-scale with values $\leq$ 1e-5 clipped to 0.

#### 4.1.3 TRAINING

All the experiments are conducted on 8 NVIDIA A100 GPUs with the AdamW (Loshchilov & Hutter, 2019) optimizer and the Exponential decay Scheduler. SSL pre-training is trained for 1M steps with $\beta_1 = 0.9$, $\beta_2 = 0.98$, a learning rate of 0.005, and a weight decay of 0.01 following [15]; ALT fine-tuning is trained for 100k steps and a learning rate of 0.0001 with other hyperparameters remaining default following [16]. All the vocoder models are trained for around 1.5M steps with $\beta_1 = 0.8$, $\beta_2 = 0.99$, an initial learning rate of 0.0001, and a weight decay of 0.9999996 following [17]. All the SVC models are trained for around 0.5M steps using $\beta_1 = 0.5$, $\beta_2 = 0.99$, and an initial learning rate of 0.0001 with 100000 decay steps following [18] and [19].

---

[13]https://huggingface.co/m-a-p/MERT-v0
[14]https://huggingface.co/facebook/w2v-bert-2.0
[15]https://github.com/khanld/Wav2vec2-Pretraining
[16]https://github.com/khanld/ASR-Wav2vec-Finetune
[17]https://github.com/NVIDIA/BigVGAN
[18]https://github.com/CNChTu/Diffusion-SVC/tree/old_Zero-Shot
[19]https://github.com/MoonInTheRiver/DiffSinger

Table 2: Low-resource ALT results of Wav2vec2 models trained, fine-tuned, and transfer-learned on different datasets. Dataset scales are annotated in hours after "-". The best result is **bold**.

| System | Unlabelled Train Data | Labelled Fine-tune Data | Labelled Transfer-learning Data | WER ($\downarrow$) |
|---|---|---|---|---|
| Whisper | / | / | / | 8.82% |
| Wav2vec2-Base | Librispeech-960 | Librispeech-960 | / | 61.16% |
| Wav2vec2-Large | LibriVox-60k | Librispeech-960 | / | 60.77% |
| Wav2vec2-Large | LibriVox-60k | Librispeech-960 | SingingVoice-10 | 7.79% |
| Wav2vec2-Large | LibriVox-60k & SingingVoice-3500 | SingingVoice-10 | / | **6.76%** |

### 4.1.4 CONFIGURATIONS

We use Wav2vec2 (Baevski et al., 2020), BigVGAN (Lee et al., 2023), NSF-HiFiGAN (Liu et al., 2022a), and DiffSVC (Liu et al., 2021) as our baseline models. The implementation details are:

- **Wav2vec2** - The original version of the Wav2vec2. We implement it using the transformers (Wolf et al., 2019), the hyperparameters and pre-trained models are adopted from [20].
- **BigVGAN** - The V2 version of BigVGAN. We implement it with pre-trained models from [17] with the same hyperparameters.
- **NSF-HiFiGAN** - The integration of NSF and HiFi-GAN, the SOTA vocoders for singing voice (Huang et al., 2023). We reimplement it using [19] with the same hyperparameters.
- **DiffSVC** - We reimplement the DiffSVC model with [19] and adopted the MRTE-based (Jiang et al., 2024) zero-shot version from [18].

### 4.2 EVALUATION METRICS

### 4.2.1 OBJECTIVE EVALUATION

We use objective metrics focusing on intelligibility, spectrogram reconstruction, F0 accuracy, and similarity with parselmouth (Jadoul et al., 2018) as the pitch extractor. We use the Amphion (Zhang et al., 2024) system for computation. The details are listed below:

- **WER** (Word Error Rate): We compute WER between the transcription and the ground truth.
- **CER** (Character Error Rate): We compute CER between the synthesized audio's transcription based on the Whisper (Radford et al., 2023) medium model and the ground truth.
- **MCD** (Mel-Cepstral Distance) (Kubichek, 1993): The distance between the synthesized audio and the ground truth audio's mel-cepstral, which shows the quality of the spectrogram reconstruction. We employ the pymcd [21] package for computation.
- **FPC** (F0 Pearson Correlation Coefficient): The Pearson Correlation of F0 trajectories.
- **F0RMSE** (F0 Root Mean Square Error): The RMSE of the log-scale F0 in cent scale.
- **SIM** (Speaker Similarity): The similarity between the converted singing voice and the target singer computed by the WavLM (Chen et al., 2022) speaker embedding model.

### 4.2.2 SUBJECTIVE EVALUATION

We use the Mean Opinion Score (MOS) and the Similarity Mean Opinion Score (SMOS) Tests for subjective evaluation. In each MOS or SMOS test, a total of 10 utterances will be evaluated. Listeners were asked to give a naturalness or similarity score between 1 and 5 with a step of 0.5 for each utterance synthesized by different systems. The ground truth audio will be provided in the MOS test, while the source and reference audio will be provided in the SMOS test. 20 volunteers who are experienced in the audio generation area are invited to the evaluation, resulting in each system being graded 200 times. The system design details in subjective evaluation are illustrated in Sec. B.

---

[20]https://huggingface.co/facebook
[21]https://github.com/chenqi008/pymcd

Table 3: Copy synthesis results on the Studio Recording and In-the-wild test settings on BigVGAN and NSF-HiFiGAN with different training sets. Dataset scales are annotated in hours after "-". The best result in each setting is **bold**. The MOS scores are within 95% Confidence Interval (CI).

| Test Data | System | Training Data | MCD ($\downarrow$) | FPC ($\uparrow$) | F0RMSE ($\downarrow$) | MOS ($\uparrow$) |
|---|---|---|---|---|---|---|
| **Studio Recording** | | Ground Truth | 0.000 | 1.000 | 0.000 | $4.31 \pm 0.23$ |
| | BigVGAN | Large-Compilation | 1.777 | **0.984** | **35.897** | $2.90 \pm 0.27$ |
| | | Large-Compilation & SingingVoice-3500 | **1.520** | 0.982 | 37.125 | $\mathbf{3.15 \pm 0.27}$ |
| | NSF-HiFiGAN | SingingVoice-165 | 2.669 | 0.932 | 83.955 | $4.13 \pm 0.25$ |
| | | SingingVoice-3500 | **2.321** | **0.962** | **59.817** | $\mathbf{4.14 \pm 0.21}$ |
| **In-the-wild** | | Ground Truth | 0.000 | 1.000 | 0.000 | $3.64 \pm 0.17$ |
| | BigVGAN | Large-Compilation | 1.700 | **0.984** | 30.931 | $3.13 \pm 0.16$ |
| | | Large-Compilation & SingingVoice-3500 | **1.420** | 0.983 | **30.503** | $\mathbf{3.55 \pm 0.15}$ |
| | NSF-HiFiGAN | SingingVoice-165 | 3.107 | 0.961 | 54.641 | $3.39 \pm 0.17$ |
| | | SingingVoice-3500 | **2.164** | **0.977** | **32.253** | $\mathbf{3.52 \pm 0.15}$ |

## 4.3 Automatic Lyric Transcription

To verify the effectiveness of large-scale singing voice data, we conduct SSL pre-training, ASR and ALT fine-tuning, and transfer-learning on different data distributions regarding Wav2vec2 (Baevski et al., 2020) models. Compared with the previous SOTA Ou et al. (2022) method on Wav2vec2, which heavily relied on transfer learning from a speech-pre-trained model, we pre-train and open-source the first Wav2vec2 model on large-scale singing voices based on our collected data, and show that we can directly tune such a model on ALT without needing extra fine-tuning and transfer learning. We use the Librispeech (Panayotov et al., 2015) and LibriVox (Kearns, 2014) datasets for speech pre-training and ASR fine-tuning. We manually sample 10 hours of high-quality annotated English singing voice for low-resource ALT. We use SingStyle111 as the test set. The transcription results are illustrated in Table. 2 with the reference accuracy provided by the Whisper (Radford et al., 2023) medium model.

It can be observed that: (1) The systems trained with purely speech data cannot handle the singing voice data, resulting in high WER values; (2) The system pre-trained and fine-tuned on speech ASR can be adapted on ALT via transfer-learning with a relatively small WER value, confirming the effectiveness of the previous work (Ou et al., 2022); (3) The system pre-trained with the singing voice can be directly tuned on ALT without transfer-learning while having a better performance, indicating the effectiveness of large-scale singing voice.

## 4.4 Neural Vocoder

We conduct vocoder training on different data distributions to verify the effectiveness of large-scale singing voice data. We pre-train and open-source SOTA BigVGAN and NSF-HiFiGAN models for singing voice applications, using their experiment results as the benchmark. The Large-Compilation distribution contains tens of thousands of speech and general sound audio mixtures following Lee et al. (2023). The SingingVoice-165 distribution contains 165 hours of high-quality studio-recorded singing voice data manually sampled by the OpenVPI team (Openvpi, 2024). The evaluation results of different systems are illustrated in Table. 3.

It can be observed that: (1) The BigVGAN trained on Large-Compilation cannot handle singing voice correctly, resulting in audio with severe glitch problems (Wu et al., 2022), significantly outperformed by the one trained on large-scale singing voice in both test settings; (2) The NSF-HiFiGAN trained on large-scale singing voice data holds a similar performance in the Studio Recording test setting and a significantly better result in the In-the-wild test setting, confirming the effectiveness of adding large-scale in-the-wild singing voice data; (3) The BigVGAN trained on Large-Compilation and our singing voice data performs best in the In-the-wild test setting, indicating the generalization ability brought by the speech and general sound.

Table 4: Singing voice conversion results on Studio Recording and In-the-wild test settings with training data in different scales. Dataset scales are annotated in hours after "-". The best result of each column is **bold**. The MOS scores are within 95% Confidence Interval (CI).

| Test Setting | Train Data | FPC (↑) | F0RMSE (↓) | CER (↓) | SIM (↑) | MOS (↑) | SMOS (↑) |
|---|---|---|---|---|---|---|---|
| **Studio Recording** | Ground Truth | / | / | 11.47% | / | / | / |
| | SingingVoice-35 | 0.894 | 119.487 | 16.56% | **0.826** | 3.72 ± 0.17 | 3.55 ± 0.25 |
| | SingingVoice-350 | **0.898** | **113.657** | 17.13% | 0.820 | 3.51 ± 0.17 | 3.30 ± 0.22 |
| | SingingVoice-3500 | 0.897 | 115.112 | **16.55**% | 0.826 | **3.73 ± 0.19** | **3.71 ± 0.27** |
| **In-the-wild** | Ground Truth | / | / | 16.52% | / | / | / |
| | SingingVoice-35 | **0.935** | **81.371** | **22.40**% | 0.744 | 3.16 ± 0.17 | 2.98 ± 0.24 |
| | SingingVoice-350 | 0.931 | 84.631 | 23.58% | 0.743 | 3.01 ± 0.17 | 2.78 ± 0.24 |
| | SingingVoice-3500 | 0.933 | 82.017 | 22.88% | **0.746** | **3.16 ± 0.16** | **3.11 ± 0.25** |

## 4.5 SINGING VOICE CONVERSION

We train our SVC model on different data scales to build experimental benchmarks for Zero-Shot SVC models. The subset of 35 and 350 hours are sampled randomly from the 3500-hour mixture, holding the same data distribution. Two evaluation settings are considered: (1) **Studio Recording Setting**: We use all the clean vocals from our SingStyle111 test subset as the source audio, and eight singers (M1, M2, M3, M4, F1, F2, F3, F4) as the target singers; (2) **In-the-wild Setting**: We use all the in-the-wild singing voice data from our SingNet test set as the source audio. We manually choose one Chinese female, one English female, one Chinese male, and one English male as four target singers. For each source utterance, we randomly sample an utterance from each target singer as the reference audio to conduct conversion. The results are illustrated in Table. 4.

It is observed that: (1) Regarding F0-related metrics, all three systems have similar performances, which meets our expectations since F0 prediction is not the bottleneck in SVC application; (2) Regarding intelligibility and similarity objective metrics, the systems trained with 35 and 3500 hours of data performances similarly, while a degradation exists on the system trained with 350 hours of data; (3) Regarding quality and similarity subjective results, the system trained with 3500 hours performs the best, while the system trained with 350 hours performs the worst.

To validate the evaluation results, we manually reviewed all synthesized samples outputted by the three systems, finding that: (1) The system trained with 35 hours of data can synthesize singing voice with accurate lyric and timbre with limited expressiveness; (2) The system trained with 350 hours of data has a better expressiveness, but the intelligibility and timbre are inaccurate with slurred words, resulting in severe quality degradation. We speculate this is because of the content and speaker encoders' inability to model many different languages and singer identities (With the increase in data scale, the languages and singer identities become more diversified, but not enough for the encoders actually to learn); (3) The system trained with 3500 hours of data alleviated the intelligibility and timbre inaccuracy with better expressiveness, as illustrated by the increase in both subjective and objective metrics, indicating the effectiveness of the large-scale data (With the rise in data scale, the encoders successfully learn the different languages and timbres). Representative cases regarding these findings can be found on our demo page[12].

## 5 CONCLUSION

This paper presents SingNet, an extensive, multilingual, and diverse Singing Voice Dataset. We collect around 3000 hours of singing voice data with various singers, languages, and styles via our proposed data processing pipeline that can extract ready-to-use training data from in-the-wild sample packs and songs online. To facilitate the use and show the effectiveness of SingNet, we pre-train and open-source SOTA Wav2vec2, BigVGAN, and NSF-HiFiGAN models on large-scale singing voices, which significantly outperform the existing open-sourced ones. We also conduct benchmark experiments on ALT, Neural Vocoder, and SVC to provide a reference for subsequent research.

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

# A    SEMI-SUPERVISED AUDIO ANNOTATION SYSTEM

## A.1    ANNOTATION WEBSITE

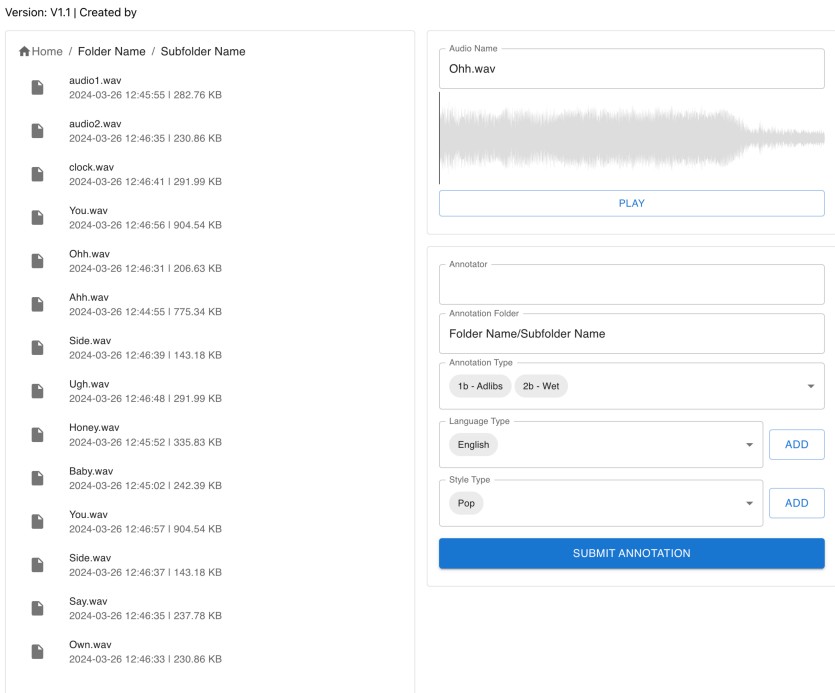

Figure 5: An overview of the audio annotation website. The sample packs used in annotation, the annotator, and the author of the annotation system are all made to be anonymous.

The audio annotation website is illustrated in Fig. 5. Annotators are asked to give folder-level annotations since most sample packs put samples of the same type into the same folder. Annotation Types are Acapella, Adlibs, Elements, FX, Dry, and Wet. Annotators should choose one label from Acapella, Adlibs, Elements, and FX, and one from Dry and Wet. Language and style annotations are also annotated for statistical results and future works.

## A.2    AUDIO CLASSIFICATION

Table 5: Audio classification accuracy results for different sample types in Music Production.

| Mixed | Acapella | Adlibs | Elements | FX |
|---|---|---|---|---|
| Dry | 86.54% | 90.38% | 66.67% | 62.50% |
| Wet | 75.76% | 73.97% | 62.07% | 75.47% |

We pre-trained a Wav2vec2 large model on the Singing Voice-3500 data mixture. We fine-tuned it on the audio classification downstream task using SingNet-SP to obtain the automatic classification model for further scaling up. The classification accuracy results are illustrated in Table. 5. It can be observed that: (1) Our model can accurately distinguish dry and wet Acapella and Adlib sounds, making it an ideal classifier since most valuable singing utterances are in these two categories; (2) Our model can distinguish dry and wet Elements and FX sounds with an accuracy around 65%, since most element and FX sounds will be filtered in the later stage, that accuracy is acceptable.

## B  SUBJECTIVE EVALUATION DETAILS

# Vocoder MOS Test

### Page 1/10

Listen to the following audios. Grade the refenrece audio first and then rate the remaining audio against the reference audio. Please evaluate the **MOS Score** on a scale of 1 to 5. 1 means poor audio quality and 5 means great audio quality.

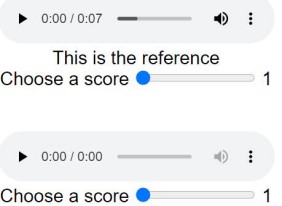

Figure 6: An overview of the MOS test.

# SVC SMOS Test

### Page 1/10

Listen to the following set of audios and rate their **Similarity** against the singer in the reference audio. The source singing audio is also provided. Please evaluate the **SMOS Score** on a scale of 1 to 5 (1: Not the same singer, 2: Slightly Not the same singer, 3: Cannot tell, 4: Slightly the same singer, 5: The same singer).

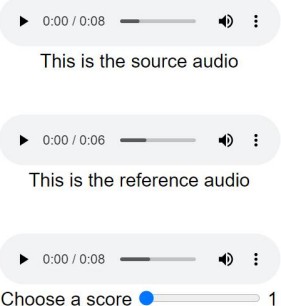

Figure 7: An overview of the SMOS test.

We adapted some ideas from the MUSHRA test to ensure the effectiveness of our subjective evaluation, as illustrated in Fig. 6 and Fig. 7. In the MOS and SMOS tests, the ground truth audio and the source audio are provided respectively for reference to avoid the bias brought by the difference in ground truth and source audio quality.

