# OpenReview forum: "SingNet: Towards a Large-Scale, Diverse, and In-the-Wild Singing Voice Dataset"
_ICLR.cc/2025/Conference — ICLR 2025 Conference Withdrawn Submission_

### Official Review · Reviewer_Fr52 · 2024-10-15

**Soundness:** 3
**Presentation:** 2
**Contribution:** 2
**Rating:** 3
**Confidence:** 4

**Summary:**

In this paper, the authors present SingNet, a 3000-hour large-scale in-the-wild singing voice dataset that's magnitudes larger than the current state-of-art singing voice dataset. The authors train self-supervised representation, vocoders and singing voice conversion models based on this collected dataset. Results demonstrate that training or finetuning with this newly proposed dataset shows improvement compared to using speech dataset to train/finetune models. The authors open-source model weights and the data processing pipeline.

**Strengths:**

This work conducts experiments in singing voice processing at an unprecedented scale, offering the research community valuable insights into model scalability and potential performance improvements.

**Weaknesses:**

The paper's primary limitations revolve around dataset accessibility and the comprehensiveness of the experimental work. While large-scale datasets are invaluable for research communities, the copyrighted nature of this dataset (88.7% from commercial sample packs, the rest from copyrighted music) prevents its release. This significantly diminishes the novelty claim of the dataset itself, as its inaccessibility hinders broader research impact.

The experiments lack sufficient breadth and depth in several areas.
- In transcription, the comparison between speech and singing voice pre-training on Wav2Vec2 shows only a 1% absolute difference after fine-tuning. Fine-tuning experiments on a more capable model, Whisper, are absent.
- In vocoding, the comparison methodology is inconsistent. While the authors fine-tune BigVGAN with the proposed dataset, they train NSF-HiFiGAN from scratch with both the OpenVPI and proposed datasets. A more balanced comparison would include fine-tuning experiments with the OpenVPI dataset as well.
- In voice conversion, the intriguing observation that the 35-hour subset outperforms the full 3500-hour set on some objective metrics warrants deeper investigation and discussion. No such is provided. This leads to doubts on the dataset quality distribution.

Given the dataset's closed nature and the questionable results from pre-trained models, the open-source data processing pipeline emerges as the paper's most potentially novel contribution. More detailed documentation on this is necessary, especially considering the potential high costs associated with reproducing the pipeline due to the use of multiple commercial plugins.

To improve the paper, the authors should provide a more detailed analysis of the data processing pipeline, including potential alternatives to commercial plugins for broader reproducibility. The experimental section should be expanded to include fine-tuning experiments on Whisper for transcription, a consistent comparison methodology for vocoding, and an in-depth analysis of the performance discrepancy between the subset and full dataset in voice conversion. Additionally, the authors should discuss potential ways to make the dataset or a subset of it accessible to the research community, considering copyright constraints.

A minor comment: please check spelling and uppercase/lowercase usage. Several examples include: Mert (Page 7) should be MERT, Wav2vec2 (throughout) should be Wav2Vec2, MERT is misspelled as MRTE on Page 8, Openvpi/OpenVPI is not used consistently throughout.

**Questions:**

It would be valuable to see experiments on fine-tuning Whisper, comparing the results of using 10 hours of data versus the proposed dataset. Similarly, including fine-tuning experiments for BigVGAN with the OpenVPI dataset would provide a more comprehensive comparison in the vocoding section.

Regarding the data processing pipeline, which appears to be a central contribution, more details would be crucial for a proper assessment of its novelty. Specifically, it would be helpful to understand how this pipeline differs from previous speech processing pipelines and why these differences constitute a novel contribution to the field. Given the importance of open resources in advancing research, are there any plans to open-source the semi-supervised captioning model mentioned in the paper?

**Details Of Ethics Concerns:**

The proposed dataset in this paper largely contains copyrighted materials. 88.7% of the duration is copyrighted commericial sample pack vocal, the rest is copyrighted music content. No ethic claims have been made in this paper that attempts to address this. Are the singers and copyright holders consenting to this research?

Additionally, the proposed vocoder models have potential to generate more powerful deepfakes and cause harm, as demonstrated in recent events. I encourage authors to either test if current deepfake detection models can reliably detect them, consider releasing their own detection models alongside the vocoders, or add watermark to the vocoder models and release watermark detection tools.

---

### Official Review · Reviewer_3Q98 · 2024-10-27

**Soundness:** 2
**Presentation:** 2
**Contribution:** 3
**Rating:** 5
**Confidence:** 3

**Summary:**

The paper presents SingNet, a large-scale and diverse singing voice dataset aimed at overcoming the limitations of existing datasets for Singing Voice Synthesis (SVS) and Singing Voice Conversion (SVC). The authors introduce a data processing pipeline that extracts high-quality singing voice data from online sources, resulting in approximately 3000 hours of multilingual and multi-style singing voices. They benchmark various state-of-the-art (SOTA) models on this dataset, including Wav2vec2, BigVGAN, and NSF-HiFiGAN, for tasks like Automatic Lyric Transcription (ALT), Neural Vocoding, and Singing Voice Conversion (SVC). SingNet's dataset and models are open-sourced to encourage further research in the field.

**Strengths:**

This paper is pioneering in providing a data processing pipeline for in-the-wild singing voice data, offering valuable insights for the singing and music community, particularly in scaling up model data. Although the dataset is still relatively small compared to speech data, it has the potential to significantly encourage the growth of singing voice data in the future.

Originality: The development of an open-source data processing pipeline for extracting singing voice data from online sources is a remarkable innovation. This approach has led to the creation of the largest in-the-wild singing voice dataset to date, broadening the horizon of research and applications in Singing Voice Synthesis (SVS) and Singing Voice Conversion (SVC).

Significance: SingNet's contribution to the field of singing voice synthesis and conversion is considerable, addressing the shortage of large-scale, diverse datasets. The open-source release of the dataset and pre-trained models is expected to expedite research and development within the audio and music generation community.

**Weaknesses:**

1. Lack of Innovation: The paper closely resembles another paper's process, Emilia, without clearly articulating its improvements over Emilia. Additionally, the paper primarily leverages existing work from others, with minimal novel technical contributions. The methods related to neural vocoding and data augmentation could benefit from more comprehensive referencing to acknowledge the contributions of earlier studies.

2. The data processing pipeline is crucial but is not described with sufficient clarity or detail, and no rationale is provided for the choice of tools, making it unconvincing. In-the-wild data, not recorded in controlled studio environments, may still contain noise, reverberation, and other audio quality issues after source separation and audio restoration, especially for more complex vocal segments. A more detailed explanation of the data processing methods is needed.

3. Although the paper compares the dataset with existing ones in terms of scale and diversity, it falls short in directly comparing its performance with other existing datasets. For instance, while the paper demonstrates SingNet's effectiveness in tasks like singing voice synthesis, it lacks direct comparisons with similar large-scale datasets on the same tasks, making it difficult for readers to ascertain SingNet's advantages in practical applications.

4. The reliability of the data processing results is questionable. The paper mentions multiple singing voice data processing pipelines but does not provide any intermediate audio demos, making it impossible to determine whether the pipeline truly yields high-quality audio. The assumption that certain modules or pre-processing techniques directly enhance performance is not thoroughly tested or validated. Furthermore, the remaining quantity of data after cleaning is not specified, making it unclear how much data is retained at each step.

5. The SingNet dataset contains a multitude of languages, It is unclear whether the language distinctions are entirely accurate, and what the error rates might be. This aspect is not detailed, yet for voice, especially singing voice, misjudging a song in one language as another can introduce bias in training.

6. The vocoder comparison for singing voices should ideally use singing data exclusively: The Large-Compilation data used in the vocoder comparison experiment primarily consists of speech data, which should be replaced with a compilation of previously published singing voice datasets for a more reasonable comparison.

**Questions:**

Questions 1 to 6 are summaries derived from Weaknesses 1-6, while 7 to 11 are additional issues:
1. Does the paper clearly articulate its improvements over Emilia?
2. What is the data processing pipeline?
3. Does SingNet lack direct comparisons with similar large-scale datasets on the same tasks?
4. Are the data processing results truly effective?
5. How is language discrimination handled?
6. Should the Large-Compilation data used in the vocoder comparison experiment be replaced with a compilation of previously published singing voice datasets for a more reasonable comparison?
7. In 3.1.1 SINGNET-SS CONSTRUCTION:
   (1) Which source separation model is used, demucs or another? The authors should clarify and provide the GitHub link for the open-source model used in the paper, rather than having readers search through related papers. Additionally, with so many source separation models available, the authors should briefly explain why they chose this particular version.
   (2) Audio Restoration: Why is FabFilter-Pro Q3 used twice? Is the order of plugins sequential or parallel? The paper does not clarify this part, causing confusion.

8. Why did the authors choose BigVGAN over subsequent Vocos for vocoder training? Only using BigVGAN and NSF-HiFiGAN as benchmark neural vocoders limits the exploration of potentially more advanced models, constraining the generalizability of the conclusions. Although the paper conducts multiple experiments, including neural vocoding, singing voice conversion, and automatic lyric transcription tasks, the experiments mainly focus on a few models and metrics.

9. Why doesn't the author fine-tune Whisper with Labelled Fine-tune data in ALT? Wouldn't the fine-tuning effect of Whisper be better?

10. How does the dataset's diversity in languages and styles affect the model's generalization capabilities? Are there specific cases where the dataset's diversity does not lead to improved model performance?

11. Could the authors discuss any limitations or challenges faced in the data extraction and cleaning process from in-the-wild sources? Are there specific types of audio data where the pipeline struggles to maintain high-quality output?

---

### Official Review · Reviewer_V71u · 2024-11-02

**Soundness:** 3
**Presentation:** 3
**Contribution:** 3
**Rating:** 6
**Confidence:** 3

**Summary:**

This paper presents a large-scale singing dataset crawled from the web and processed with multiple techniques, along with its processing methods. It significantly surpasses previous work in terms of data volume, language coverage, and style variety. The authors validate the effectiveness of their dataset across various benchmarks.

**Strengths:**

* The proposed dataset surpasses previous one in both quality and diversity, facilitating related research significantly.
* The authors validate the effectiveness of their dataset across multiple tasks, enhancing the persuasive value of its application.

**Weaknesses:**

* The statistics of the dataset can be more fine-grained, such as the distribution of singer gender, the pitch distribution of each gender, and the sub-distributions of each language and style.
* Despite possible difficulties of doing this on large datasets, the lack of fine-grained MIDI and phone duration annotations makes it relatively challenging for applying this dataset to singing voice synthesis (SVS) tasks. Some MIDI notation and alignment models may help the annotation.

**Questions:**

* Did the authors evaluate the impact of different processes in their pipeline on the quality of the singing voice and the performance of downstream tasks? For example, do dry and wet recordings influence the performance of downstream tasks?

**Details Of Ethics Concerns:**

Since the original data is crawled from the web, there may be copyright issues. The authors should clarify the relevant copyright concerns and usage permissions.

---

### Official Review · Reviewer_Va21 · 2024-11-02

**Soundness:** 2
**Presentation:** 2
**Contribution:** 2
**Rating:** 3
**Confidence:** 5

**Summary:**

The paper proposes SingNet, a comprehensive and diverse dataset for singing voices. The authors address the limitations in singing voice applications like Singing Voice Synthesis (SVS) and Singing Voice Conversion (SVC), by compiling 3000 hours of singing voices from various languages and styles. The dataset is created through a data processing pipeline that extracts clean data from sample packs and internet songs. SingNet's utility is proven using pre-trained and state-of-the-art (SOTA) models like Wav2vec2, BigVGAN, and NSF-HiFiGAN. Benchmark experiments include Automatic Lyric Transcription (ALT), Neural Vocoder, and Singing Voice Conversion (SVC). Compared to existing datasets, SingNet offers significantly more diversity, making it a crucial resource for large-scale, data-driven singing voice applications.

**Strengths:**

1. The authors claim to release the first and largest diverse dataset for singing voice applications. If they eventually open-source the data and processing code, it will be a tremendous contribution to the academic community.
2. The authors propose a data preprocessing pipeline designed for in-the-wild data, addressing important issues such as audio quality reconstruction, data filtering, etc.

**Weaknesses:**

1. The authors claim that the proposed dataset can boost the development of singing voice synthesis (SVS). However, the common definition of SVS is synthesizing singing voices from lyrics and melody annotations (such as MIDIs or F0s). There is no evidence or verification in this paper to support that the proposed dataset supports such SVS tasks.
2. A similar concern, in lines 33-35, the authors claim that the way ACEStudio creates singing voices is manpower-consuming and inconvenient for scaling up, because it requires tuning the in-detailed pitch, phoneme, and duration information. Even so, ACEStudio can generate singing voices from textual annotations. I can't see how the proposed dataset can be used to train a model that generates singing voices without such tedious tuning.
3. The comparison with current datasets is somehow unfair. Datasets, like NHSS, CSD, M4Singer, Opencpop, etc., consist of detailed annotations, such as words, phonemes, word/phoneme durations, MIDI sequences, F0 sequences, etc., which prevents the scaling up of the data. However, the authors did not propose a new automated method to accelerate this process or reduce its cost. Instead, they chose to bypass it altogether; that is, their preprocessing method does not include any text alignment processes like MFA, or MIDI extraction processes. This results in their dataset containing only audio, limiting their experiments to ASR (which further needs Whisper as the rater), vocoder, and SVC.
4. Even though the simplicity of the proposed dataset restricts the authors to conducting only semi-supervised or audio-only experiments, they are still not comprehensive enough. For example, for audio-only reconstruction or compression methods, the authors only investigate vocoders, such as BigVGAN and NSF-HiFiGAN. However, with the advancement of voice generation methods with language modeling techniques, discrete representations like audio codecs are also an important part of acoustic representation. This paper lacks evidence of the effectiveness of their data on codec tokenizers.
5. From the perspective of an audio-only dataset, this paper also lacks many statistical conclusions. For example, how many singers are in this dataset? What is the gender structure? What is the age structure? What is the voice part structure?

In conclusion, this paper lacks significant contributions or contains overstated contributions. From a methodological perspective, the data preprocessing approach is simple, being a direct application of various SOTA methods (not even applied in a new domain); from a dataset perspective, although this dataset is currently the largest, it lacks many of the basic annotations required for SVS and is missing crucial statistical information.

**Questions:**

1. The authors claim that they trimmed the leading and trailing silences of each segment. However, the samples on their demo web page still consist of noticeable silences (for example, the very first sample consists of a leading silence of 7 seconds), why?
2. In line 245, the authors mentioned "the time stamps in the human-labeled lyrics". Is there any detail about this? Are all the lyrics labeled by humans?
3. The authors build benchmarks in ASR, vocoder, and SVC, why do they choose to only open-source the checkpoints of ASR and vocoder, not SVC? Is there any specific reason?

**Details Of Ethics Concerns:**

No concern.

---

### Note · Authors · 2024-11-22

I have read and agree with the venue's withdrawal policy on behalf of myself and my co-authors.